# Enhanced bulk photovoltaic effect in two-dimensional ferroelectric $CuInP_2S_6$

Yue Li[1,2,5], Jun Fu[1,2,5], Xiaoyu Mao[1,2,5], Chen Chen[1,2], Heng Liu[1,2], Ming Gong[3,4✉] & Hualing Zeng [1,2✉]

The photocurrent generation in photovoltaics relies essentially on the interface of p-n junction or Schottky barrier with the photoelectric efficiency constrained by the Shockley-Queisser limit. The recent progress has shown a promising route to surpass this limit via the bulk photovoltaic effect for crystals without inversion symmetry. Here we report the bulk photovoltaic effect in two-dimensional ferroelectric $CuInP_2S_6$ with enhanced photocurrent density by two orders of magnitude higher than conventional bulk ferroelectric perovskite oxides. The bulk photovoltaic effect is inherently associated to the room-temperature polar ordering in two-dimensional $CuInP_2S_6$. We also demonstrate a crossover from two-dimensional to three-dimensional bulk photovoltaic effect with the observation of a dramatic decrease in photocurrent density when the thickness of the two-dimensional material exceeds the free path length at around 40 nm. This work spotlights the potential application of ultrathin two-dimensional ferroelectric materials for the third-generation photovoltaic cells.

[1] International Center for Quantum Design of Functional Materials (ICQD), Hefei National Laboratory for Physical Science at the Microscale, and Synergetic Innovation Center of Quantum Information and Quantum Physics, University of Science and Technology of China, 230026 Hefei, Anhui, People's Republic of China. [2] Key Laboratory of Strongly-Coupled Quantum Matter Physics, Chinese Academy of Sciences, Department of Physics, University of Science and Technology of China, 230026 Hefei, Anhui, People's Republic of China. [3] CAS Key Laboratory of Quantum Information, University of Science and Technology of China, 230026 Hefei, People's Republic of China. [4] Synergetic Innovation Center of Quantum Information and Quantum Physics, University of Science and Technology of China, 230026 Hefei, Anhui, People's Republic of China. [5]These authors contributed equally: Yue Li, Jun Fu, Xiaoyu Mao. ✉email: gongm@ustc.edu.cn; hlzeng@ustc.edu.cn

The bulk photovoltaic effect (BPVE), a kind of nonlinear optical process that converts light into electricity in solids, has a potential advantage in a solar cell with an efficiency that exceeds the fundamental Schockley–Queisser (S–Q) limit[1–7]. This effect is only valid in crystals with broken inversion symmetry, which can lead to significant electronic polarization[8–10] that plays a similar role as the p–n junction and Schottky barrier in photovoltaics. Thus, a strong zero-bias photocurrent (or shift current) can be observed in noncentrosymmetric materials when irradiated with light above the energy bandgap[11,12]. The BPVE is particularly pronounced in polar materials[13–15]. The recent advances of the BPVE in ferroelectric perovskite oxides exceeding the S–Q limit have spurred further exploration for electronic systems with optimal device efficiency performance[16]. Along this line, the low-dimensional materials such as $CuInP_2S_6$, $\alpha$-$In_2Se_3$, and $MoTe_2$ with layered van der Waals (vdW) structure without inversion symmetry have been suggested for high-efficiency photocurrent collection via BPVE[17–23]. However, while the intriguing physics in the two-dimensional (2D) limit, such as magnetism[24–26], hyperferroelectricity[27–30], and exotic topological phases[31–33] have been widely explored, the direct experimental study on BPVE in 2D materials remains unimplemented.

Here we demonstrate the enhanced BPVE in ultrathin ferroelectric $CuInP_2S_6$ (CIPS). The shift current is directly measured in the family of 2D materials. The enhanced BPVE is associated to the room-temperature ferroelectricity of $CuInP_2S_6$ as evidenced by the electric polarization, temperature, and thickness dependent behavior. Below the critical temperature, $T_c$ ~315 K[17,34] for the transition from ferroelectric phase to paraelectric phase, large open-circuit photovoltage ($V_{oc}$), and short-circuit photocurrent ($I_{sc}$) are observed along the direction of electric polarization when irradiated with light. The $V_{oc}$ and $I_{sc}$ can be controlled and even reversed by an external electric field. Based on this approach, the maximum $V_{oc}$ is obtained to be about 1.0 V and the enhanced photocurrent density is two orders of magnitude higher than that reported from the ferroelectric perovskite oxides[35–39]. Notably, above $T_c$, negligible photovoltage and photocurrent are observed in all devices in the paraelectric phase with restored high symmetry[40]. We find that the shift current density from 2D CIPS falls between that from one-dimensional (1D) and three-dimensional (3D) BPVE-based photovoltaics. We also demonstrate a crossover from 2D to 3D photovoltaics when the thickness of the CIPS exceeds the free path length ($l_0$) of photocarriers, beyond which the photocurrent reduces to the dark condition level even with inversion asymmetry. These results highlight the potential of developing next-generation solar cells with ultrathin 2D materials.

## Results

**BPVE in 2D ferroelectric CIPS**. In previous investigations of the BPVE on bulk materials, a capacitor-like structure is utilized (see Fig. 1a), where the dielectric layer or polar material is sandwiched by the top and bottom metal electrodes[35,36,39]. When it is irradiated with light, a non-zero $I_{sc}$ could be observed even without external bias. This zero-bias photocurrent is regarded as the characteristic feature of BPVE. The BPVE has been mostly demonstrated in perovskite oxides in the name of photoferroics[3]. For example, a giant photovoltage above the bandgap was observed in domain engineered $BiFeO_3$[3,5], and the power conversion efficiency $\eta$ possibly above the S–Q limit was reported[16]. In these experiments, the sample thickness is much larger than the free path length of hot photocarriers, which is about 10–100 nm in a rough estimation[36,41]. In a 1D nanotube with a large free path length, the photocurrent can be furtherly enhanced. Therefore, ultrathin vdW ferroelectrics with

controllable thickness are ideal candidates for enhanced photocurrent generation via BPVE (see Fig. 1a).

In this work, 2D CIPS was utilized as the photoferroic layer for the BPVE study. The CIPS is an ionic ferroelectric, where the Cu, In, and paired P–P atoms fill in the sulfur cages. The off-center shift of the Cu cations leads to stable out-of-plane ferroelectricity[17,42]. The electrically switchable polarization in CIPS has been demonstrated at room temperature down to the thickness of 4 nm[17]. Figure 1b presents the optical image of the 2D photovoltaic device with a sandwiched structure (for device #1). All the ultrathin vdW materials used in the devices were prepared by mechanical exfoliation method[43] and assembled via the dry-transfer technique[44] (see the detailed fabrication process in the "Methods" section). The ultrathin CIPS with thickness down to 8 nm was confirmed by the atomic force microscopy (AFM) and the Raman spectroscopy (see Fig. S1) and their ferroelectricity was confirmed by the piezoresponse force microscopy (PFM) measurement at ambient condition (see Fig. S2). In the device, bilayer and few-layer graphene were used as crossed bottom and top electrodes, respectively. Benefited from the high optical transmission of graphene in the visible range[45,46], our device is allowed to carry out the direct measurements on photo-generated carriers with the synchronous electric control of the polarization.

The CIPS is a semiconductor with a gap width $E_g$ at ~2.9 eV[47,48], thus to realize the BPVE, the laser with a wavelength of 405 nm is chosen as the excitation source. We scan the laser spot (with a diameter of 2 μm, see the "Methods" section) over the device at its initial state. From the $I_{sc}$ mapping, we find that the photo-generated current exists only in the overlapping area of the top and bottom graphene electrodes as evidenced by the spatial correspondence (see Fig. 1b, c). In contrast, outside the overlapping sandwich area, the photocurrent is negligible at around 2 pA, which is to the level of the background current as limited by the DC source-monitor (see the "Methods" section). To check the spatial dependence of $I_{sc}$ in detail, we extracted one line profile from the $I_{sc}$ mapping (see Fig. S3). When the laser spot is moved away from the sandwich edge with step resolution at 160 nm, the $I_{sc}$ vanishes quickly to the noise level. This spatial confinement feature is a direct manifestation of finite free path length of the hot photocarriers. This feature excludes the origin of $I_{sc}$ from the Schottky barrier at the single or double graphene/ CIPS interfaces. Moreover, the sign and magnitude of the photocurrent are non-uniformly distributed, showing independence on sample topography (see Figs. 1c and S3). Since the device is not electrically polarized in advance, the spatially randomized photocurrent is the result of the spontaneous electric polarization in CIPS (see Fig. S2).

Next, we studied the photocurrent generation in CIPS by homogeneous light illumination on the whole device. In the open-circuit condition, we characterize the relation between photocurrent and voltage using[16]

$$V_{oc} = \frac{J_{sc}d}{\sigma_{pv} + \sigma_d},\qquad(1)$$

where $d$ is the sample thickness, $J_{sc}$ is the photocurrent density from the total photocurrent $I_{sc}$ divided by the laser spot area, and $\sigma_{pv}, \sigma_d$ (with $\sigma_{pv} \gg \sigma_d$) is the conductivity of the photovoltaic current and the dark current, respectively. In theory $\sigma_{pv} = eI_0\alpha\varphi(\hbar\omega)^{-1}\mu\tau$, where $I_0$ is the light intensity, $\alpha$ is the absorption coefficient, $\hbar\omega$ is the excitation photon energy, $\varphi$ is the quantum yield and $\mu, \tau$ are the mobility and lifetime of the hot carriers. The photocurrent density is related to the BPVE via the

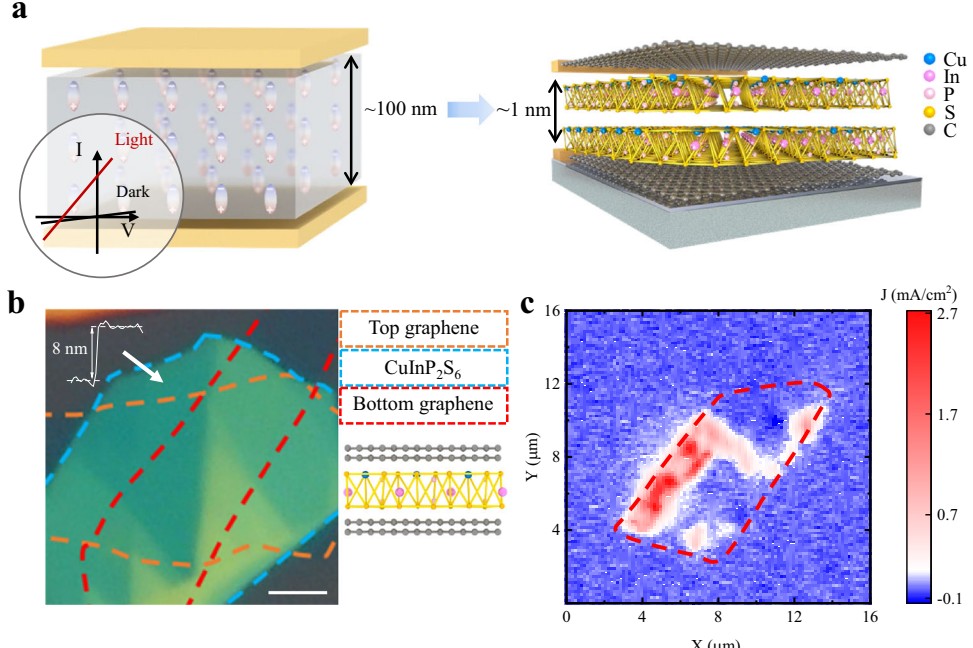

**Fig. 1 Bulk photovoltaic effect in 2D ferroelectric CIPS. a** (Left panel) The schematic structure of 3D BPVE device with film thickness at the order of around 100 nm. Inset shows the characteristic *I–V* curve of the bulk photovoltaic effect in 3D ferroelectrics. The black and red lines represent the *I–V* curves at the dark conditions and bright conditions, respectively. (Right panel) The schematic structure of 2D BPVE device. The thickness of 2D ferroelectrics can be as thin as 1 nm. **b, c** The optical image and corresponding short-circuit photocurrent density mapping of the 2D BPVE device. The region of top graphene, ultrathin CIPS, and bottom graphene are marked with orange, blue, and red dashed lines, respectively. The scale bar in **b** is 3 μm.

third-rank piezoelectric tensor as

$$J_{sc} = \alpha g I_0 \propto I_0, \qquad (2)$$

This relation can be understood from two different aspects by defining the equivalent resistance $R = d/(\sigma_{pv} + \sigma_d)$ and built-in photovoltaic electric field $E_{pv} = J_{sc}/(\sigma_{pv} + \sigma_d)$. Intuitively, a device with high efficiency is required to have large equivalence resistance $R$ and a large photovaltavic field $E_{pv}$, which are essential for photocurrent generation instead of electron–hole recombination. Thus, the larger the photovoltaic field is, the stronger the photovoltaic voltage will be.

**Switchable photovoltaic effect in 2D BPVE device**. Figure 2a shows the characteristic output *I–V* curve from one typical device (device #2). Due to the atomically thin thickness of 2D ferro-electrics, the bias in the DC measurement is set to be within ±0.1 V (much smaller than $V_{oc}$), which is equivalent to a field strength of $E_m \sim 10^5$ V/cm, to avoid the degradation of electric polarization in ultrathin CIPS (see Supplementary Note 1). With light illumination, a large and positive $I_{sc}$ was observed. Using the laser spot area ~20 μm$^2$, we estimate the short-circuit current density $J_{sc}$ to be around 10 mA/cm$^2$. In contrast, when the laser is switched off (dark condition), the current density at 0.1 V is about 1 μA/cm$^2 \ll J_{sc}$. With a linear fit on the output characteristic *I–V* curve, the $V_{oc}$ is estimated to be −0.8 V, which is close to the coercive field of CIPS (see Fig. S2). Compared with previously reported bulk ferroelectric photovoltaic cells[35–37], this result yields much larger $I_{sc}$ and $V_{oc}$ at the same excitation density (10.2 W/cm$^2$). These data yield photoconductivity at the order of $8.7 \times 10^{-7}$ S/m, which is comparable with the results reported in literatures[35–37]. A full characterization of the ferroelectrics for BPVE is summarized in Table 1, highlighting the potential advantages of CIPS for BPVE based photovoltaics by comparing them with the other materials in the literature.

In ferroelectrics, the electric polarization is nonvolatile and electrically switchable. Therefore, the BPVE of CIPS should be in principle controllable via an external electric field. To verify the relation between the enhanced photocurrent and the ferroelectricity, we study the photocurrent as a function of poling voltage. Figure 2b presents the photovoltaic behavior of the device (for device #3) with the pulsed poling voltage varying from +1.8 to −1.8 V and then backward to +1.8 V. When polarizing the device with +1.8 V, a positive photocurrent is observed within ±0.1 V reading bias. With −1.8 V poling voltage, the current is modulated to negative. By scanning the polarizing voltage, the sign of the photocurrent can be controlled from negative to positive gradually. The asymmetry of the photocurrent between backward and forward scanning arises from the ferroelectric hysteresis of CIPS. Specifically, the switchable photocurrent is summarized in Fig. 2c, where the BPVE can be enhanced or suppressed subject to the electric polarization state of CIPS. To show the hysteresis behavior more clearly, we extract the $J_{sc}$ at each poling voltage. As shown in Fig. 2d, the $J_{sc}$ follows an obvious hysteresis loop when the poling voltage scans forward and backward. Similarly, the $V_{oc}$ as the function of the poling voltage is shown in Fig. S5.

**Temperature and power dependence of the BPVE in CIPS**. To further investigate the origin of BPVE in CIPS, we study the BPVE as a function of temperature (*T*). CIPS is a room temperature ferroelectric with critical temperature $T_c$ at ~315 K[17,34]. Upon heating the device to be above $T_c$, the ferroelectric CIPS will transform to the paraelectric phase with restored inversion symmetry, in which the BPVE is forbidden strictly. Figure 3a shows the open-circuit voltage as a function of temperature, in which the $V_{oc}$ decreases with the increase of temperature and vanishes completely when $T > 315$ K. The disappearance of the photovoltage above $T_c$ indicates that the observed enhanced BPVE is associated with the ferroelectricity of 2D CIPS. We also

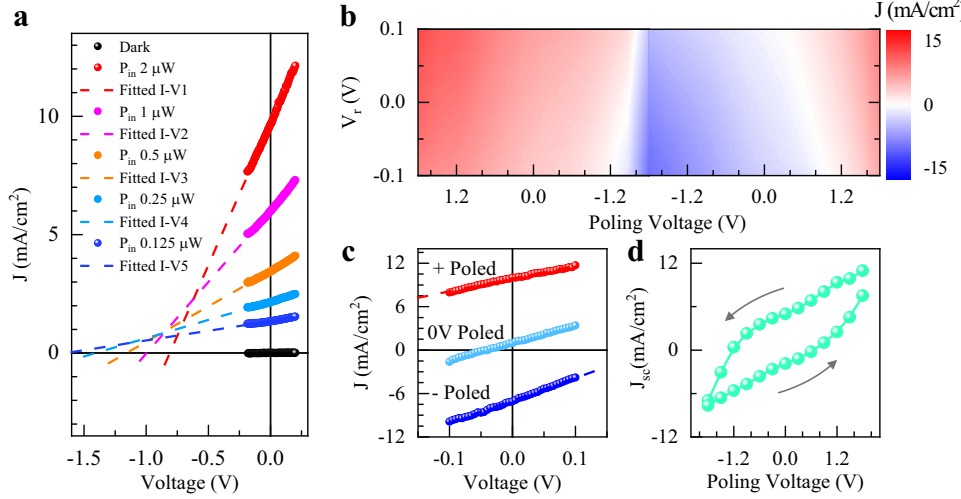

**Fig. 2 Switchable photovoltaic effect in 2D BPVE device. a** The characteristic output I–V curves of device #2 at dark (black dots) and bright (color dots) conditions. The excitation light source is the laser line with 405 nm wavelength. The dotted color lines represent the linear fit on the output I–V curves measured at bright conditions with different laser power. **b** The photovoltaic behavior under different poling voltages. The poling voltage is swept from +1.8 to −1.8 V and then backward to +1.8 V. The reading bias is set to be within ±0.1 V. The color represents the magnitude of the generated photocurrent density. **c** Extracted output I–V curves at specific poling voltages from **b** with the red, blue, and dark blue dots representing the positively, zero voltage, and negatively poled states respectively. **d** The plot of $I_{sc}$ as a function of the poling voltage. The arrows indicate the measurement sequence.

**Table 1 The results of short-circuit current density $J_{sc}$, open-circuit voltage $V_{oc}$, photoconductivity $\sigma$, fill factor (FF), and power conversion efficiency $\eta$ for samples with thickness $d$ and excitation intensity $P_{in}$.**

| | $J_{sc}$ (A/cm²) | $V_{oc}$ (V) | $d$ (nm) | $P_{in}$ (W/cm²) | $\sigma$ (S/m) | FF (%) | Gap (eV) | S-Q limit (%) | $\eta$ (%) |
|---|---|---|---|---|---|---|---|---|---|
| BTO tip enhanced[16] | $1.9 \times 10^{-2}$ | 1.2 | 100 | 0.1 | $1.6 \times 10^{-5}$ | 21 | 3.2 | 2.8 | 4.8 |
| BTO 20 nm[36] | $2.2 \times 10^{-6}$ | 0.6 | 20 | 0.3 | $7.3 \times 10^{-10}$ | 25 | 3.2 | 2.8 | $1.1 \times 10^{-4}$ |
| BTO 50 nm[36] | $3.8 \times 10^{-6}$ | 0.65 | 50 | 0.3 | $2.9 \times 10^{-9}$ | 25 | 3.2 | 2.8 | $2.1 \times 10^{-4}$ |
| BFO[35] | $3.8 \times 10^{-7}$ | 0.27 | 170 | $7.5 \times 10^{-4}$ | $2.4 \times 10^{-9}$ | 25 | 2.7 | 8.8 | $3.4 \times 10^{-3}$ |
| KBNNO[37] | $4 \times 10^{-8}$ | 3.5 | $2 \times 10^4$ | $4 \times 10^{-3}$ | $2.3 \times 10^{-9}$ | 31.5 | 1.39 | 32.9 | $1.1 \times 10^{-3}$ |
| PLZTN[38] | $3.7 \times 10^{-8}$ | 23 | $3 \times 10^5$ | 0.1 | $4.8 \times 10^{-9}$ | 46 | 2.5 | 12.5 | $3.9 \times 10^{-4}$ |
| BBLT[39] | $6.5 \times 10^{-9}$ | 16 | $5 \times 10^5$ | 0.1 | $2.03 \times 10^{-9}$ | 29 | 3.2 | 2.8 | $3.0 \times 10^{-5}$ |
| CIPS | $9.6 \times 10^{-3}$ | 0.8 | 7.3 | 10.2 | $8.71 \times 10^{-7}$ | 25 | 2.85 | 7.1 | 0.63 (/$\alpha$ ~ 3%) |
| CIPS | $8 \times 10^{-4}$ | 1.65 | 7.3 | 0.3 | $3.3 \times 10^{-8}$ | 25 | 2.85 | 7.1 | 3.67 (/$\alpha$ ~ 3%) |
| CIPS | $2.6 \times 10^{-5}$ | 0.5 | 230 | 25.5 | $1.17 \times 10^{-7}$ | 25 | 2.85 | 7.1 | $5.1 \times 10^{-5}$ (/$\alpha$ ~ 25%) |

Data in the first seven rows are from previous literature; the last three rows are data from our samples. The photoconductivity per power density can be obtained using $\sigma/P_{in}$, in which the CIPS almost have the smallest conductivity per excitation density. For the power conversion efficiency, the best value for thin CIPS films in this study is closing to the S-Q limit when considering the absorption at 405 nm (see Supplementary Note 2).

investigate the photocurrent dependence on light illumination power $P_{in}$. Figure 3b displays the $J_{sc}$ and the corresponding $\sigma_{pv}$ from Eqs. (1) and (2) as a function of the laser power (for device #2) changing from 1 to 20 W/cm². The monotonical increase of generated $J_{sc}$ with $P_{in}$ is consistent with the prediction of Eq. (2) and the results in previous studies of BPVE[21,35,36]. By the extracted open-circuit voltage, we also extract $\sigma_{pv}$, which is proportional to the excitation power as expected from Eqs. (1) and (2). By comparing the photoconductivity of CIPS with that from previous literatures[31–35], we find comparable results in our devices. Importantly, noting that in all the experiments, our samples have the smallest thickness with a similar amplitude of $V_{oc}$ and $\sigma_{pv}$, thus the CIPS has the smallest photoconductivity per excitation power if using the data in Table 1.

**Energy band alignments in the 2D CIPS device.** The observed enhanced photocurrent via BPVE in CIPS can be understood

through the switchable energy band alignments in the graphene/CIPS/graphene heterostructure. In the ferroelectric phase, the off-center shift of the Cu ions induces a built-in electric field ($E_{bi}$) perpendicular to the 2D plane. Therefore, with an out-of-plane electric polarization, the vacuum levels at the two sides of the CIPS are different as illustrated in Fig. 4a, b, which leads to different alignments of the energy bands of graphene at the top and bottom interfaces. The different energy band alignments result in an asymmetric energy potential barrier with barrier height at $\Delta\Phi$ along the out-of-plane direction. For ferroelectric CIPS, the $\Delta\Phi$ could be as large as ~1 eV[49]. When the device is illuminated with light, the photo-generated electron and hole pairs are separated by the $E_{bi}$ and are drifted under $\Delta\Phi$ to generate significant $I_{sc}$. Furthermore, the sign of the asymmetric energy potential barrier between the top and bottom surfaces can be controlled by the direction of electric polarization, leading to the direction switchable photocurrent (Fig. 2c). When the CIPS is with the paraelectric phase, the charged Cu ions are equally

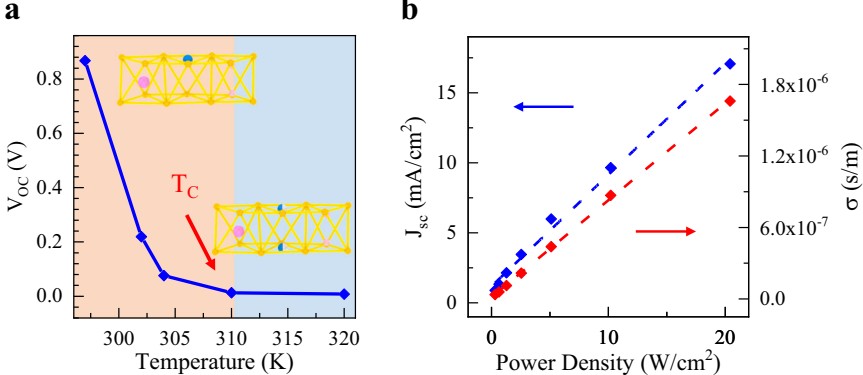

**Fig. 3 Temperature and power dependence of the BPVE in CIPS. a** The open-circuit voltage as a function of the temperature. The $V_{oc}$ vanishes when the temperature increases to the phase transition temperature at about 315 K. Insets display the crystal structures of CIPS in ferroelectric and paraelectric phases. **b** The short-circuit current density (left axis) and the photoconductivity (right axis) as a function of the illumination laser power density. Linear dependence of the $J_{sc}$ and $\sigma_{pv}$ on excitation power is in accord with Eq. (2).

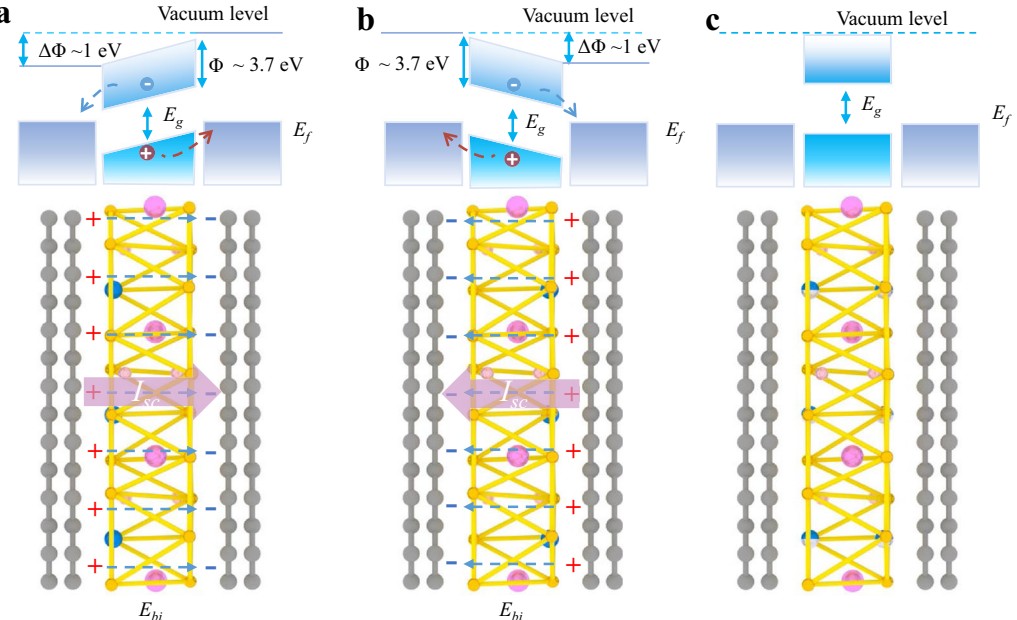

**Fig. 4 Energy band alignments in the 2D CIPS device. a, b** The band diagram for the graphene/CIPS/graphene heterostructure at the upward and downward electric polarization state of ferroelectric CIPS. The top and bottom graphene electrodes are represented with slabs of atoms in gray on the left and right sides of CIPS. In the heterostructure, the polarization charges are marked with "+" and "−" symbols, and the built-in electric field are indicated by the dashed lined arrows. The cartoon on the top depicts the corresponding energy shift of the conduction band and valence band. **c** The band diagram of the device with the CIPS at the paraelectric phase.

distributed at the two surfaces in the unit cell, resulting in a zero $E_{bi}$ and $\Delta\Phi$ as shown in Fig. 4c. There is no observable photocurrent from BPVE.

**The dimensionality effect of the BPVE**. Finally, to show the advantages of 2D BPVE, we summarize the photocurrent density generated via BPVE from different material systems in Fig. 5a. By considering the linear dependence of the photocurrent on light power density from Eq. (2), we take three colored shadings to represent the performance for 1D, 2D, and 3D BPVE photovoltaics, respectively. We find that the BPVE induced photocurrent density in 2D CIPS falls in the gap between 1D (WS₂ nanotube in ref. [21]) and conventional 3D ferroelectric photovoltaics. For 3D materials, most of the ferroelectrics are oxide insulators with perovskite structure, in which the large gap width

fundamentally limits the photocurrent density unless via subtle domain engineering (refs. [3,5,6]) or under the aid of scanning probe technique (ref. [16]). In 1D WS₂ nanotube, the ultrahigh photocurrent density is benefited from the nanoscale diameter size of the tube in nature and the enhanced shift current[50]. The BPVE-induced photocurrent density is three orders of magnitude larger than that observed in 3D bulk photovoltaics. However, as confined by the tube structure, the direction of the photocurrent could not be switched by the external electric field. For 2D ferroelectric CIPS, the bulk photovoltaic performance of all the devices (with thickness ranging from 8 to 60 nm) fill in this three orders of magnitude gap in $J_{sc}$. We have also studied 2D CIPS with various thicknesses (see Fig. 5b), showing that when the thickness is $<d_c \sim 2l_0$, where $l_0 \sim 40$ nm, significantly enhanced BPVE can be found. In contrast, when the film thickness is larger than the above free path length, the photocurrent decreases to the

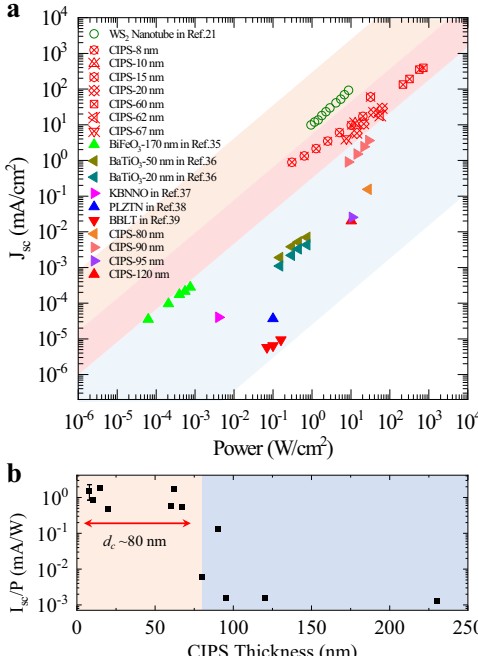

**Fig. 5 The dimensionality factor of the BPVE. a** An overview of the photovoltaic performance in 1D, 2D, and 3D BVPE materials. The paralleled polygons filled in yellow, red, and blue represent the short-circuit photocurrent density from 1D, 2D, and 3D bulk photovoltaics respectively. With the same illumination power density, the photocurrent density from 2D ferroelectric CIPS falls in the gap between 1D and 3D bulk photovoltaics. **b** The thickness dependent BPVE in ferroelectric CIPS. For each device with different film thicknesses, the observed short-circuit photocurrent is normalized by the incident light power. The $I_{sc}$ decreases to the 3D noise level as the CIPS is thicker than 80 nm, indicating the free path length of the hot photocarriers to be about 40 nm, using the estimation of $d_c \approx 2l_0$. A similar conclusion can be achieved by the open circuit voltage, as shown in the Supplementary material of Fig. S10.

noise level quickly. This is a clear demonstration of the crossover from 2D to 3D BPVE materials, which has not yet been observed in experiments. A similar conclusion can be achieved by examining the $V_{oc}$ (see Fig. S10). Our results suggest that the dimensionality of the photovoltaic materials plays an essential role in photo-electric conversion. The observed dimensionality effect in BPVE photovoltaics can be intuitively interpreted from two aspects. Firstly, it is well-known that the separation between the two collecting electrodes in a photovoltaic device should be smaller than the free path length for efficient photoelectric conversion (see the Cartoon in Fig. S13). With a capacitor-like device structure, the atomic thickness of 2D CIPS naturally fulfills this prerequisite. Secondly, it has been pointed out that when the dimensionality is reduced the BPVE is more efficient with enhanced shift current[51], which may originate from the Berry connection of the Bloch functions[52].

## Discussion
In summary, we report the enhanced BPVE effect in 2D ferroelectric CIPS. In the ferroelectric phase without inversion symmetry, the photocurrent is enhanced by two orders of magnitude as compared with that in bulk ferroelectric materials. In contrast, in the paraelectric phase with inversion symmetry, the BPVE is absent even in the 2D limit. The performance of our 2D photovoltaics falls in between the 1D and 3D bulk photovoltaics, implying that the device dimensionality is one of the key ingredients for developing high-efficiency BPVE-based photovoltaics.

From the thickness-dependent photocurrent generation, we estimate the free path length in CIPS to be about 40 nm. Our findings highlight the potential of ultrathin 2D ferroelectrics for developing third-generation solar cells with high efficiency beyond the fundamental S–Q limit.

## Methods
**Sample preparation and topography measurement**. Bulk single crystal graphite and CIPS used in the study were purchased from 2D Semiconductors Inc. The ultrathin vdW materials were prepared by the mechanical exfoliation on Si substrate with 300 nm SiO₂ coated on the top. The vdW heterostructures were made by the all-dry transfer technique[42]. The topography and thickness of samples were characterized by atomic force microscopy (AFM) (AIST-Smart SPM) at tapping mode.

**Raman spectroscopy characterization**. Raman spectra were carried out with the Horiba micro-Raman system (LabRAM HR Evolution) to verify the phase structure and thickness of CIPS and graphene. The wavelength of the excitation laser is 532 nm and the on-sample power is set to be 150 μW. As shown in Fig. S1, the characteristic Raman mode of single crystal CIPS at 275, 325, 384 cm⁻¹ were observed, which confirm the ferroelectric phase and are consistent with the previous reports[16]. The graphene thickness is confirmed by the optical contrast under an optical microscope and the Raman spectra. For a few layers of graphene, the intensity ratio of G mode and 2D mode can be used to identify film thickness. The ratio of $I_G/I_{2D}$ for the bottom graphene electrode in device #1 is 2/3, which confirms that it is bilayer graphene.

**PFM characterization**. PFM measurements were performed with the AIST-Smart SPM system under ambient conditions. The PFM tip used is Pt/Ir coated conducting Si tip. The conductive substrates used in this study were prepared through sputtering 20 nm-thick gold film onto the Silicon substrate. Ultrathin CIPS was mechanically cleaved directly on the conductive substrate. In the PFM measurements, the piezoelectric amplitude and phase show obvious butterfly-like loop and hysteresis loop as shown in Fig. S2, respectively.

**Device fabrication and transport measurement**. The metal electrodes on the top and bottom graphene were made by the standard UV lithography technique and the following electron beam evaporation of 5 nm Ti and 60 nm Au. The samples were soaked in acetone for 12 h during the lift-off process. The transport measurement was taken in the vacuum chamber under the pressure blow 0.1 Torr at room temperature and above room temperature. Keysight source meter (B2912A with 10 fA resolution) was used to apply pulsed voltage and measure the I–V characteristics. The excitation laser wavelength for the photovoltaic measurement is 405 nm. The laser was focused on the sample with a homemade scanning confocal microscopy system.

## Data availability
The data that support the findings of this study are available from the corresponding author on reasonable request.

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

## Acknowledgements
This work was supported by the National Key Research and Development Program of China (Grant Nos. 2017YFA0205004, 2018YFA0306600, 2017YFA0304504 and 2017YFA0304103), the National Natural Science Foundation of China (Grant Nos. 11674295 and 11774328), the Fundamental Research Funds for the Central Universities (Grant Nos. WK3510000013 and WK2030020032), and the Anhui Initiative in Quantum Information Technologies (Grant No. AHY170000). This work was partially carried out at the USTC Center for Micro and Nanoscale Research and Fabrication.

## Author contributions
H.Z. conceived the idea and supervised the research. Y.L., J.F., and X.M. prepared the samples, fabricated the devices, and carried out the photocurrent measurements, H.L. and C.C. carried out the PFM measurements. Y.L., J.F., X.M., M.G., and H.Z. analyzed the data and wrote the paper. All authors commented on the manuscript.

## Competing interests
The authors declare no competing interests.
