## [Peer Review File · Nature Communications]

REVIEWER COMMENTS

Reviewer #1 (Remarks to the Author):

This paper demonstrates an approach to fabricate ferroelectric CuInP2S6-based photovoltaic cell. The device performance is enhanced owing to the ferroelectricity of the 2D CuInP2S6. The experiment design and analysis are systematic. However, there are several important points that require further clarification, as summarized below.

1. The reason for using CuInP2S6 and ferroelectric materials as the acting layer is not clear. In fact, there are numerous reported work of photovoltaic cells using traditional materials and other 2D materials for the wavelength of 405 nm, and the devices exhibit better performance, stability, and yield than this work (<https://doi.org/10.1016/j.rser.2018.06.031>). Besides, the ferroelectricity of the CuInP2S6 is not stable and will disappear when the temperature is above 315 K. Therefore, the practical use of such device is limited. The authors need to provide a more convincing reason of using this material.
2. Based on our experience, the thin and large CuInP2S6 flake is difficult to achieve simultaneously. Thus, the calculated value (normalized value) of J_{sc} of the device with thickness of 7.3 nm is overestimated and is still much lower than the value of traditional materials. It would be better to provide a more comprehensive benchmarking including a wider range of materials rather than just list the ferroelectric materials-based photovoltaic cell.
3. For the PFM measurement, the phase hysteresis direction is different from the previous work (DOI: 10.1038/ncomms12357), could the author explain this discrepancy?
4. It has been mentioned that the coercive voltage of CuInP2S6 is around 0.8 V. However, in the Figure 2a, it shows the ferroelectric enhanced photocurrent when a very small voltage is applied. Could the author explain more about this phenomenon?
5. For Figure 5b, more detailed data (I_{sc} vs V under different laser power of the samples with different thickness) needs to be provided instead of only plotting the thickness identification in the supplementary information. The summarized data without individual device characterization result is not convincing.
6. In fact, temperature dependent ferroelectricity of CIPS has been reported in previous work (DOI: 10.1038/ncomms12357), which renders this work lacks of novelty.
7. The efficiency of the photovoltaic cell is critical and should be provided. Meanwhile, it would be better to provide a thorough comparison with other representative work.
8. There are many typos in the main text. For example, it should be "photocurrent" instead of "photocorrent" in page 3. Besides, references 16 and 47 are over-replicated.

Reviewer #2 (Remarks to the Author):

Two dimensional ferroelectric materials are of great interests from many aspects, including but not limiting to memories, transistors. Zeng and Gong et al. reported here the investigation of bulk photovoltaic effect in 2D CIPS. They provide clear evidence of the polarization dependence on the

photovoltage, and also the presence of a critical thickness, which are quite interesting and could be published in nature commun. Some changes and further study are recommended.

1. The critical concern is that the prospect of such bulk photovoltaic effect in photovoltaic, such as solar cells, is questionable, given the poor current density achieved and the significant temperature effect (Figure 3a). Instead, providing the photodetection behavior, including switching speed, responsivity, could help to understand its potential in other related fields.
2. What is the wavelength spectral of the BPVE in CIPS?
3. A model and fitting to experiment is necessary to properly interpret the effect of free path length in Figure 5b.
4. Again in Figure 5b, is it possible that the thickness dependence is simply conductance effect, which scale inversely with thickness?
5. A major conclusion is that the dimensionality plays essential role in the photoelectric conversion, whereas the underlying mechanism is not elucidated.
6. In Figure 3a, a full temperature range is suggested to investigate the bulk photovoltage and its current density.
7. The authors should also elaborate the thickness effect on Voc, in addition to current density. Does increasing thickness result in higher Voc?

Reviewer #3 (Remarks to the Author):

In this manuscript, the authors reported the bulk photovoltaic effect in two-dimensional ferroelectric CuInP2S6 with enhanced photocurrent density by two orders of magnitude higher than conventional bulk ferroelectric perovskite oxides. These results are interesting for CuInP2S6 which can be potentially used for developing next generation solar cells with ultrathin 2D materials. However, before the paper can be accepted, some necessary comments are listed below.

1. The authors emphasize that the ultrathin vdW ferroelectrics are ideal candidates for enhanced photocurrent generation via BPVE, why? Can authors give a detailed theoretical analysis?
2. Why the mapping data in Figure 1c looks so scatter? Is it caused by the uneven thickness of CIPS?
3. Why the authors used the fitted data in Figure 2a, but not use the experimental data?
4. The open circuit voltage data over different temperature is very scattered, as shown in Figure 3a, especially for the data near 310 K. More data are needed.
5. The thickness of CIPS presented in Figure 5b are shown in Figure S8, but those height lines can't be used as the evidence. The authors should provide the detailed AFM images.
6. For the devices, the thickness of the crystal is not uniform, such as in Figure 1b, Figure S4a, which can cause different current distribution in the device.
7. Some small concerns. (1) The scale bars should added into the optical images, such as Figure 1b, Figure S4a, Figure S6a, Figure S7a and b. (2) The authors used the Keysight source meter B2902A to carry out the I-V measurement, the resolution of normal B2902A is 100 fA. The authors achieved a 2 pA level measurement, how did the author achieve electromagnetic shielding?

Reviewer #4 (Remarks to the Author):

In this manuscript, Li et al. investigated the bulk photovoltaic effect (BPVE) in 2D ferroelectric CuInP2S6. A significant enhanced photocurrent density compared to conventional bulk ferroelectric perovskite oxides was demonstrated. The BPVE was attributed to the ferroelectric polarization in the 2D CuInP2S6. The topic is of current interest and the results are well presented. The manuscript can

be considered for publication provided that several questions can be better addressed.

1. How could the spatial confinement exclude the origin of I_{sc} from the Schottky barrier considering the vertical structure?
2. Did the CIPS materials with different thicknesses have the same bandgap of ~ 2.9 eV, did they see any thickness dependence in the studied range? By the way, the quoted references gave the values of 2.4 and 2.9 eV for thicker or bulk samples.
3. Generally, for the BPVE, the V_{oc} varies linearly with the thickness d . This correlation could be provided.
4. In Fig 5b, the normalized I_{sc} values have a sharp drop when the thickness is larger than 80 nm. Does this observation mean the generated carriers collected by the electrodes have a sudden drop when the thickness is larger than 80 nm? Why a sudden change instead of a smooth one is observed, more physical explanation could be provided.
5. As known, the BPVE will generate a large photocurrent density in small-thickness materials with a relatively low open-circuit voltage. Considering the practical application, is there any method to overcome this shortcoming?
6. Recent reviews on 2D Ferroelectrics, such as *Adv. Electron. Mater.* 2020, 6, 1900818; *Adv. Mater.* 2021, 33, 2005098, could be included, which might be useful to the broad readership to better understand this emerging topic.
7. The manuscript is in general well written. However, a few typos should be corrected, such as Page 2 "The enhanced BPVE is associated to".

Table of Contents

1. Point-by-Point response to review report #1 (pages 2 - 9)
2. Point-by-Point response to review report #2 (pages 9 - 15)
3. Point-by-Point response to review report #3 (pages 15 - 23)
4. Point-by-Point response to review report #4 (pages 23 - 28)
5. Major changes to the manuscript (pages 29 - 30).

Point-by-Point response to Review Report #1

We thank the Referee #1 for his/her insightful suggestions and positive comments that “The experiment design and analysis are systematic”. Below, we have addressed all the issues raised in this review report. A summary of the major changes can be found in the last section of this file.

Question 1: The reason for using CuInP_2S_6 and ferroelectric materials as the acting layer is not clear. In fact, there are numerous reported works of photovoltaic cells using traditional materials and other 2D materials for the wavelength of 405 nm, and the devices exhibit better performance, stability, and yield than this work (<https://doi.org/10.1016/j.rser.2018.06.031>). Besides, the ferroelectricity of the CuInP_2S_6 is not stable and will disappear when the temperature is above 315 K. Therefore, the practical use of such device is limited. The authors need to provide a more convincing reason of using this material.

Response: We appreciate the reviewer for his/her concern about adopting the CIPS as the acting layer. The four major reasons are summarized below, in which the first and the second viewpoints have been presented in the introduction part of the manuscript.

1. CIPS is a new 2D ferroelectric material realized in recent years. While the ferroelectricity has been widely explored in 3D bulk materials, it is rare in nature for 2D materials. The direct experimental study on BPVE in 2D ferroelectrics remains unimplemented.
2. BPVE is a nonlinear optical effect widely exists in materials without inversion symmetry. The ferroelectric photovoltaics via BPVE could generate a giant photovoltage larger than the band gap width, and can produce power conversion efficiency above the Shockley-Queisser limit based on the P-N junction photovoltaics.
3. The critical temperature of this material from the ferroelectric phase to the paraelectric phase is well above the room temperature, which may be used for practical applications.
4. From the fundamental physics point of view, it is necessary to study the relation between dimensionality and the photovoltaic effect, which has never been clarified clearly in experiments.

Question 2: Based on our experience, the thin and large CuInP₂S₆ flake is difficult to achieve simultaneously. Thus, the calculated value (normalized value) of J_{sc} of the device with thickness of 7.3 nm is overestimated and is still much lower than the value of traditional materials. It would be better to provide a more comprehensive benchmarking including a wider range of materials rather than just list the ferroelectric materials-based photovoltaic cell.

Response: We agree with the referee that the mechanical isolation of large and uniform ultrathin vdW films is challenging in 2D materials, especially for the 2D ferroelectrics with relatively strong vdW interaction between the neighboring layers. However, in our experiments, we can prepare large and thin CIPS flakes using an Au-coated substrate, following the method as detailed in *Nat. Commun.* *11*, 2453 (2020). By this method, uniform CIPS flakes with thickness larger than 6 nm and lateral size larger than 40 μm can be successfully realized (see more data from the optical image in **Fig. R1**). In **Fig. R1b - 1c**, the ferroelectricity is verified via the second harmonic generation (SHG) mapping and the single-point piezoresponse force microscopy (PFM) hysteresis loops.

For the data presented in the main text, we have verified that the ferroelectricity is robust in ultrathin CPIS samples with thickness greater than 6 nm with uniform lateral sizes about several microns, which is larger than the spot size of our excitation laser ($\sim 2 \mu\text{m}$ in diameter). Thus, for the calculation of J_{sc} in each device, the photocurrent density has not been overestimated. To be more specific, in our measurements, the photocarriers are only excited under the laser spot. Since the dark current of the whole device is about 3 - 4 orders of magnitude lower than the photocurrent J_{sc} , the current under irradiation is mainly contributed to the separation of the generated carriers under the laser spot. For these reasons, in the calculation of J_{sc} , the photocurrent density is normalized to the area of the laser spot, but not the area of the whole CIPS photovoltaic device. This kind treatment is also used in the previous literature. Therefore, despite that the CIPS device presented in **Fig. 1b** is not uniform with some kind of thickness fluctuation, it will not lead to overestimation of J_{sc} . The spatial dependence of J_{sc} (or J_{sc} imaging) is presented in **Fig. 1c** and **Fig. S3**. More similar data from other devices can be found in **Fig. S4** and **Fig. S11**.

Figure R1: (a) Optical image of large-area and thin layer CIPS on Au coated silicon substrate (from our *unpublished data*). Insert is the AFM line profile along the white arrow. The scale bar is 10 μm . (b) The corresponding SHG imaging of the sample in (a). The second harmonic generation (SHG) intensity indicates the sample thickness. (c) PFM measurement of the CIPS flakes in (a).

Question 3: For the PFM measurement, the phase hysteresis direction is different from the previous work (DOI: 10.1038/ncomms12357), could the author explain this discrepancy?

Response: The question raised by the referee is a technical issue in PFM measurement. Using different tip-sample resonance frequencies in PFM, it can yield different hysteresis loops with opposite directions. We show this phenomena in our data in **Fig. R2** by slightly tuning the tip measuring frequencies. In these two data, the PFM magnitudes are the same, whereas the PFM hysteresis loop direction is flipped when the frequency is changed from a lower value to a higher value (see **Fig. R2b** and **R2c**). This PFM phase hysteresis direction is consistent with the previous literature (X. Wang *et. al.*, *Nat. Commun.* 10, 3037 (2019)).

The reason for this technical issue is clarified as follows. In the PFM measurement, the phase information can represent the anti-parallel of electric polarizations in the ferroelectrics. It comes from the opposite response of a given ferroelectrics, either compressive or tensile, under an external electric field. However, for the specific compressive or tensile behavior, the response can not be simply distinguished by the PFM phase. Since the PFM involves the lockin amplifier technique, the phase information represents only a relative physical quantity. The specific correlation between the polarization state and the PFM phase is thus influenced by the PFM tip frequency, which varies from measurement to measurement.

Figure R2: (a) PFM tip resonant frequency. (b-c) PFM phase hysteresis loop with PFM tip frequencies lower (at 310 kHz) and higher (at 330 kHz) than the resonant frequency (at 322 kHz) as indicated by the two vertical dashed lines in (a).

Question 4: It has been mentioned that the coercive voltage of CuInP_2S_6 is around 0.8 V. However, in the Fig. 2a, it shows the ferroelectric enhanced photocurrent when a very small voltage is applied. Could the author explain more about this phenomenon?

Response: There are two different concepts here. One is the coercive voltage, which is required to be relatively large for the realization of ferroelectric polarization switching; and the other is the readout voltage, which is required to be much smaller than the coercive voltage to avoid the depolarization effect of the reading bias.

For 2D materials, due to the ultrathin film thickness at atomic scale, low voltage applied along the direction perpendicular to the 2D plane results in a large electric field strength. For example, for films with the thickness of 1 nm, applying 1 V DC voltage leads to an electric field strength about 1×10^7 V/cm along the out-of-plane direction. This value is much larger than the coercive electric field strength in the 2D ferroelectric CIPS ($\sim 7.5 \times 10^5$ V/cm as shown in **Fig. S2**). Therefore, applying a wide range reading bias in photocurrent study leads to a strong depolarization effect in CIPS. To avoid this problem, when measuring the I-V curve at bright state in all our devices, the reading voltage is limited to the range of ± 0.1 V (equivalent to an electric field about $\sim 10^5$ V/cm). The V_{oc} is then determined through a linear fit to the I-V curve near zero voltage (see **Fig. 2a** and **Fig. S6**).

Question 5: For Figure 5b, more detailed data (I_{sc} vs V under different laser power of the samples with different thicknesses) needs to be provided instead of only

plotting the thickness identification in the supplementary information. The summarized data without individual device characterization result is not convincing.

Response: Following this suggestion, we add the I-V curves for all the devices (see **Fig. R3**) as summarized in **Fig. 5b** in the revised supplementary information. These detailed data are also presented in **Fig. S15**

Figure R3: (a-f) The photovoltaic transport character of devices in Fig. 5 (b) for different thicknesses (in the unit of nm) and different excitation powers (in the unit of μ W). The intercept photocurrent at zero bias (for example in (a), $I_{sc} = I_{ds}(V_{ds} = 0)$ is 1.2 nA) divided by the excitation laser spot area yields the photoresponse density used in the main text.

Question 6 : In fact, temperature dependent ferroelectricity of CIPS has been reported in previous work (DOI: 10.1038/ncomms12357), which renders this work lacks of novelty .

Response: We thank the referee for comparing our results with that by Fucui Liu et al, which used the same material of thickness down to 4 nm with the critical temperature T_c about 320 K. For the T_c study, our results are consistent with this paper. However, these two papers aim to discuss totally different physics.

The work by Liu et al (*Nat. Commun.* 7,1357 (2016)) aimed to study the 2D

ferroelectricity in CIPS. However, our paper studies the bulk photovoltaic effect and its crossover from 2D to 3D in the thin CIPS films. A central point we want to demonstrate is that the BPVE in this material indeed arises from its ferroelectricity, thus when the system recovers its inversion symmetry in the paraelectric phase above T_c , the BPVE is disappeared. This is a critical issue because the similar photocurrent generation may also be realized based on the Schottky junction at the hetero-structure interfaces. We have made some more experimental measurements to reinforce this conclusion in **Fig. R4** by using the SHG measurement in a much wider temperature regime (from 90 K to 370 K).

To the best of our knowledge, this is the first work trying to measure the photovoltaic current directly in 2D ferroelectric thin films. Our results show the potential application of CIPS in the third generation photovoltaic cells. We hope this work can stimulate the searching of BPVE and the related physics in the other 2D materials.

Figure R4: Temperature-dependent zero-bias photocurrent (a) and open-circuit voltage (b) in the CIPS devices. (c) SHG as a function of temperature for samples of bulk material, thick film (60 nm), and ultrathin film (6 nm). In the ultrathin film, the phase transition is not smooth probably from the finite size effect, which is consistent with the results in Liu et al, *Nat. Commun.* 7,1357 (2016).

Question 7: The efficiency of the photovoltaic cell is critical and should be provided. Meanwhile, it would be better to provide a thorough comparison with other representative works.

Response: We thank the referee for this suggestion, which has been added in the revised manuscript (see **Table I** and **Supplementary Information Note II**).

We calculated the power conversion efficiency (PCE) using

$$\eta = \frac{J_{sc} \times V_{oc} \times FF}{P_{in} \times \alpha}, \quad (1)$$

where J_{sc} is the short-circuit current density, V_{oc} is the open-circuit voltage in the light irradiation, P_{in} is the power of the incident light, α is the absorption coefficient of 2D CIPS, and FF is the fill factor, which represents the ratio between actual power and the ideal power obtained from the product $J_{sc} \times V_{oc}$. Considering the thickness of the CIPS used in our study, the ultrathin CIPS is transparent in a broad range of the light spectrum. Most intensity of the irradiation light is not absorbed by the device. To quantify the absorbance of ultrathin CIPS at 405 nm irradiation, we measure the transmission spectrum of CIPS on PDMS substrate as shown in **Fig. R5**. We find more than 97% of the light intensity transmits through the 10 nm thick CIPS. The absorption coefficient α at 405 nm is found to be lower than 3% for 10 nm CIPS. According to the data shown in **Fig. 2a** (device #2 with $J_{sc} \sim 0.8$ mA/cm², $V_{oc} \sim 1.65$ V, and FF ~ 25 %) under on-sample light power density at 0.3 W/cm², the PCE of device #2 is estimated to be 3.67 %, which is more than half of the value at the S-Q limit for 2D CIPS with 2.85 eV band gap. We compare this efficiency with the conventional ferroelectrics based photovoltaic cells. For example, the PCE of 50 nm thin film ferroelectric BaTiO₃ is 2.1×10^{-6} . Some more data can be found in the last column of **Table I** (see the main text). Our efficiency is appealing among all materials based on BPVE.

Figure R5: The transmission spectra of 10 nm (a) and >200 nm (b) CIPS thin films measured at 405 nm. In our experiment, the 405 nm laser is adopted as the irradiation source. The transmission (T) is defined as $T = I_{CIPS}/I_{sub}$, where I_{CIPS} and I_{sub} are the transmitted light intensity of CIPS on PMDS and bare PDMS substrate. We estimate the transmission coefficient for 10 nm CIPS to be about 97%, which is used in the main text for the estimation of PCE.

Question 8: There are many typos in the main text. For example, it should be “photocurrent” instead of “photocorrent” in page 3. Besides, references 16 and 47 are over-replicated.

Response: We have improved the presentation of the whole manuscript and have fixed all the typos in the main text and the supplementary materials. The duplicated reference has been removed.

Point-by-Point response to review report #2

We thank the referee #2 for his/her suggestive and positive comments that “The results provide clear evidence of the polarization dependence on the photovoltage, the topic is interesting and could be published in Nature Communications”. As detailed below, we addressed all the issues raised and revised the manuscript accordingly.

Question 1: The critical concern is that the prospect of such bulk photovoltaic effect (BPVE) in photovoltaic, such as solar cells, is questionable, given the poor current density achieved and the significant temperature effect (Figure 3a). Instead, providing the photodetection behavior, including switching speed, responsivity, could help to understand its potential in other related fields.

Response: We thank the referee for this critical comment, which is a common issue in the study on BPVE at current stage. However, this system can still enable us to demonstrate some fundamental issues in photovoltaic effect, such as the possibility of exceeding the S-Q Limit as discussed in *Nature Photonics* **10**, 611–616 (2016). New possibilities are also promising in regarding of the tunable polarization and domain structure by an electric field, which is also one of the main driving forces for the exploring of ferroelectricity in materials without inversion symmetry.

We appreciate the suggestion on studying the switching speed and photoresponsivity of 2D CIPS, which actually are under investigation in our group. We find that the polarization switching speed is at the order of kHz in our devices, which is limited by the relatively long capacitive charging/discharging time of the CIPS samples with lateral size at $\sim 10 \mu\text{m}$. It can be further improved with the decreasing of sample sizes. The direct testing of polarization switching speed of our devices is shown in **Fig. R6a**. We use the zero-bias photocurrent at +3 V poling as a reference, and switch the polarization by sending -3 V pulses with different pulse duration. We find that the

zero-bias photocurrent can be modulated with 0.1 ms pulse width. With the pulse duration increased to 1 ms, the polarization can be fully flipped and the photocurrent is saturated. This result implies that the switching speed of the CIPS device can be faster than 1 kHz. We notice that the switching time required for our CIPS device is still longer than that demonstrated in the other ferroelectric memories. The reason is most likely due to the large device size (the overlapping area is $5 \mu\text{m} \times 10 \mu\text{m}$ size), resulting in a large capacitive effect. In a previous work studying the ferroelectric tunneling junction based on ultrathin CIPS (*Nat. Electron.* 3, 466–472 (2020)), the writing/erasing time is in the range of 10-50 μs in devices of $2 \mu\text{m} \times 2 \mu\text{m}$ sample size. In this regard, the switching speed can be further improved by the minification of the size of our CIPS devices.

For the photodetection behavior, we measure the photocurrent responsivity of another device at -1 V bias under low power light irradiation (1 nW (P)). As shown in **Fig. R6b**, the maximum photocurrent is -4 pA ($I_{\text{illuminated}}$) in the overlapped area and the dark current is -0.8 pA (I_{dark}). The photo responsivity is then given by $R = \frac{(I_{\text{illuminated}} - I_{\text{dark}})}{P} = 3.2 \text{ mA/W}$ for this device. This value is comparable with those reported in the previous literature of ferroelectric materials, which is however still much lower than the conventional photodetectors.

Figure R6: (a) The switching speed of the BPVE device. The zero-bias photocurrent at +3 V polarized state is taken as a reference. The photocurrent direction is switched with -3 V pulse voltage with different pulse duration. (b) Photocurrent mapping at -1 V bias under 1 nW laser irradiation at 405 nm.

Question 2: What is the wavelength spectral of the BPVE in CIPS?

Response: The BPVE dependent on the excitation wavelength according to previous studies. To check this, we scan the wavelength of the irradiation laser from 400 nm to 455 nm (with the aid of a wavelength-tunable laser), while keeping the same laser power. The wavelength spectral of the BPVE in 2D CIPS is shown in **Fig. R7**. We find that photons with higher energy can induced more significant BPVE, which is consistent with our discussion that the photon energy should be larger than the band gap width of CIPS for BPVE excitation. With linear function fit to the wavelength dependent photocurrent, we find that the photocurrent will decreases to the noise level when the wavelength is larger than ~ 450 nm, indicating that the energy gap of CIPS should be larger than $1240/450 = 2.76$ eV. This estimation is further confirmed in the absorption spectrum measurement of 2D CIPS, in which the energy band gap of CIPS thin films is determined to be ~ 2.83 eV.

Figure R7: The wavelength dependent BPVE in CIPS flakes. The zero-bias photocurrent is plotted as a function of laser wavelength under the same irradiation power.

Question 3: A model and fitting to experiment is necessary to properly interpret the effect of free path length in Figure 5b.

Response: The free path length may be formulated using the results in *Phys. Rev. Lett.* 118, 096601 (2017) and *Phys. Rev. B.* 90, 161409(R) (2014). It is a complicated function, depending strongly on the third-rank tensor G_{ijl} , mobility, lifetime and carrier energy etc. For the thin film CIPS, some of these parameters are still lacking. However, from the previous literature, there are some intuitive facts that can be used

to understand these results. Firstly, the thick materials lack BPVE effect even in the presence of inversion symmetry breaking; and Secondly, the 3D bulk material can be used for BPVE if and only if the thickness is small enough. For this reason, with the decreasing of thickness in ferroelectric materials, the photocurrent will be almost constant when the thickness is smaller than the free path length, and it will exponentially decrease to the noise level when the thickness is greater than the free path length. We explain this result using a cartoon in **Fig. R8**, showing that the BPVE is efficient only when $d \leq 2l_0$. This criteria is used in *Phys. Rev. Lett. 118, 096601 (2017)*, which finds that the free path length is of the order of 10 - 100 nm in the BTO thin film. Our analysis here is consistent with these results. We have modified these discussions in the revised manuscript, according to the above physical picture.

Figure R8: Illustration of the diffusion process in photocarriers created by irradiation. The carriers will be driven by the built-in electric field E_{bi} , which can reach the two electrodes simultaneously only when the distance is less than $2l_0$, with l_0 being the free path length. Here we let the electron and hole to have the same free path length. When $d > 2l_0$, the created electron and hole can not reach the electrodes efficiently, yielding a sudden drop of photocurrent.

Question 4: Again in Figure 5b, is it possible that the thickness dependence is simply conductance effect, which scales inversely with thickness?

Response: We have addressed this issue in details in **Question 3** from two different aspects, which can not be interpreted from the conductance effect. Firstly, the exponentially decreases of the photocurrent is shown in *Phys. Rev. Lett. 118, 096601 (2017)*. Secondly, it can be understood using the cartoon in **Fig. R8**, based on which we estimate the free path length l_0 .

Question 5: A major conclusion is that the dimensionality plays essential role in the photoelectric conversion, whereas the underlying mechanism is not elucidated.

Response: The reason for the dimensionality effect is detailed from two aspects in the following.

Firstly, it is well-known that in photovoltaics the separation between the two electrodes should be smaller than the free path length (or diffusion length) of photocarriers (see a cartoon illustration in **Fig. R8**). For some of the indirect band gap materials (such as silicon) with long photocarrier lifetime, this separation in devices can be large (as large as several microns). However, for the direct band gap semiconductors based photovoltaics, it is small. In most of the materials (such as perovskites, CIPS, β -Ga₂O₃, In₂O₃, and SnO₂ etc), the free path length is typically about tens of nanometers to hundreds of nanometers. Therefore, with finite free path length, the film thickness in the photovoltaic device is one of key factors to determine the photoelectric conversion efficiency, especially for BPVE based photovoltaics. For 2D materials, they are naturally with atomically thin thickness that is much smaller than the free path length of both indirect and direct band gap materials. With a capacitor-like structure, efficient BPVE is more likely to be realized in two dimensional materials if compared with bulk materials as illustrated in **Fig. 1a**.

Secondly, it has been pointed out that the BPVE is more efficient in reduced dimensionality (*Cook et al, Nat. Commn. 8, 14176 (2017)*) and in materials with smaller energy gaps (*Grinberg et al, Nature 503, 509 (2013)*). These results can be understood from the shift current, which may originate from the Berry connection of the Bloch functions when the inversion symmetry is breaking. This mechanism is generally utilized in the theoretical calculation of the BPVE (*Phys. Rev. Lett. 109, 236601 (2012)*) and recently is used to interpret the enhanced BPVE in 1D system. For example, in tungsten disulfide WS₂ nanotubes, the 1D structures result in increased density of states near the band gap energy, which is important for the enhanced shift current. For semiconductors, it is also known that when the dimensionality is reduced the electron-phonon interaction is much weaker, hence much longer carrier lifetime. This important principle has been discussed in details in the recent review article (*Liang Z. Tan et al, npj computational materials, 2, 16026 (2016)*). Therefore, for 2D CIPS with inversion symmetry breaking, there exists enhanced shift current from similar underlying mechanism.

For the above two reasons, we expect the important role of dimensionality in our study. Therefore, we clarify this role by plotting the BPVE for materials with various

dimensions in **Fig. 5a**, by summarizing the results for the three dimensional materials and one dimensional materials from the existing well-established literature. In this plot, the CIPS system can be regarded as two dimensional when the film thickness is smaller than the free path length. In the revised manuscript, some more discussions about the dimensionality effect is also added.

Question 6: In Figure 3a, a full temperature range is suggested to investigate the bulk photovoltage and its current density.

Response: We thank the referee for this constructive suggestion. We have carried out some new experiments by studying the temperature-dependent I_{sc} and V_{oc} with higher resolution, which as shown in **Fig. R9**. The zero-bias photocurrent decreases with the increasing of temperature and finally vanishes above 315 K. To check the ferroelectric to paraelectric phase transition, we carried out a temperature dependent SHG from 90 K to 370 K. For all the CIPS samples from bulk to atomically thin films, we find the critical temperature is at about 320 K. However, for 6 nm CIPS thin film, the phase transition is not as sharp as the bulk material (the similar result has been reported in *Nature Communications* 7, 12357 (2016), with the critical temperature at about 320 K). By plotting these SHG in the same figure (see **Fig. R9c**), we estimate the critical temperature for all thickness to be 320 K, which is used in the main text.

Figure R9: Temperature-dependent zero-bias photocurrent (a) and open-circuit voltage (b) of CIPS device. (c) SHG measurements as a function of temperature for samples of bulk material, thick film (60 nm), and ultrathin film (6 nm). In the ultrathin film, the transition is not smooth probably from the finite size effect, which is consistent with the results in *Nat. Commun.* 7,1357 (2016).

Question 7: The authors should also elaborate the thickness effect on V_{oc} , in addition to current density. Does increasing thickness result in higher V_{oc} ?

Response: We thank the referee for this comment. Same to the thickness dependent of I_{sc} , the similar dependence can also be found in V_{oc} as shown in **Fig. R10**, in which V_{oc} vanishes in the thick samples. We have added this result in the revised Supplementary Information as **Fig. S10**. We find that V_{oc} is significant only when the thickness is less than 80 nm. However, the drops of V_{oc} is not as sharp as the short-circuit current I_{sc} .

Figure R10: Open-circuit photovoltage of CIPS devices with various thicknesses. When the thickness is less than the free path length, the V_{oc} is independent of the thickness; however, when the thickness exceeds the free path length, V_{oc} quickly drops to zero. This observation is consistent with the BPVE theory, which is independent of sample thickness.

Point-by-Point response to review report #3

We thank the Referee #3 for his/her suggestive and positive comments that “The results are interesting for CuInP_2S_6 , which can be potentially used for developing next generation solar cells with ultrathin 2D materials. However, before the paper can be accepted, some necessary comments are listed below.” Here, we have addressed all the issues raised in this review report. A summary of the major changes can be found in the last section of this file.

Question 1: The authors emphasize that the ultrathin vdW ferroelectrics are ideal candidates for enhanced photocurrent generation via BPVE, why? Can authors give a detailed theoretical analysis?

Response: We thank the referee for raising this fundamental issue. The reason for that the ultrathin vdW ferroelectrics are ideal candidates for enhanced photocurrent generation via BPVE is detailed from two aspects in the following. Firstly, it is well-known that in photovoltaics the separation between the two electrodes should be smaller than the free path length (or diffusion length) of photocarriers (see a cartoon illustration in **Fig. R11**). For some of the indirect band gap materials (such as silicon) with long photocarrier lifetime, this separation in devices can be large (as large as several microns). However, for the direct band gap semiconductors based photovoltaics, it is small. In most of the materials (such as perovskites, CIPS, β -Ga₂O₃, In₂O₃, and SnO₂ etc), the free path length is typically about tens of nanometers to hundreds of nanometers. Therefore, with finite free path length, the film thickness in the photovoltaic device is one of key factors to determine the photoelectric conversion efficiency, especially for BPVE based photovoltaics. For 2D materials, they are naturally with atomically thin thickness that is much smaller than the free path length of both indirect and direct band gap materials. With a capacitor-like structure, efficient BPVE is more likely to be realized in two dimensional materials if compared with bulk materials as illustrated in **Fig. 1a**.

Secondly, it has been pointed out that the BPVE is more efficient in reduced dimensionality (*Cook et al, Nat. Commn. 8, 14176 (2017)*) and in materials with smaller energy gaps (*Grinberg et al, Nature 503, 509 (2013)*). These results can be understood from the shift current, which may originate from the Berry connection of the Bloch functions when the inversion symmetry is breaking. This mechanism is generally utilized in the theoretical calculation of the BPVE (*Phys. Rev. Lett. 109, 236601 (2012)*) and recently is used to interpret the enhanced BPVE in 1D system. For example, in tungsten disulfide WS₂ nanotubes, the 1D structures result in increased density of states near the band gap energy, which is important for the enhanced shift current. For semiconductors, it is also known that when the dimensionality is reduced the electron-phonon interaction is much weaker, hence much longer carrier lifetime. This important principle has been discussed in details in the recent review article (*Liang Z. Tan et al, npj computational materials, 2, 16026 (2016)*). Therefore, for 2D CIPS with inversion symmetry breaking, there exists enhanced shift current from similar underlying mechanism.

For the above two reasons, we expect that ultrathin vdW ferroelectrics with reduced dimensionality are ideal candidates for enhanced photocurrent generation via BPVE. And we also clarify this role by plotting the BPVE for materials with various dimensions in **Fig. 5a**, by summarizing the results for the three dimensional materials

and one dimensional materials from the existing well-established literature. For ultrathin CIPS, the photocurrent generation via BPVE is significant.

Figure R11: Illustration of the diffusion process in photocarriers created by irradiation. The carriers will be driven by the built-in electric field E_{bi} , which can reach the two electrodes simultaneously only when the distance is less than $2l_0$, with l_0 being the free path length. Here we let the electron and hole to have the same free path length. When $d > 2l_0$, the created electron and hole can not reach the electrodes efficiently, yielding a sudden drop of photocurrent.

Question 2: Why the mapping data in Fig. 1c looks so scatter? Is it caused by the uneven thickness of CIPS?

Response: The scatter of the photocurrent mapping data in **Fig. 1c** arises from the randomly distributed ferroelectric domains. Similar result can be found in **Fig. S3c**. To better answer this question for the possible origin of uneven thickness of CIPS, we measure the photocurrent distribution in a new device with uniform sample thickness as shown in **Fig. R12**. The mapping is taken with higher spatial resolution. We find that, similarly, the mapping data is scattered. Accordingly, the photocurrent in different positions also exhibit negative and positive values. Therefore, the scattered photocurrent data is the manifestation of ferroelectricity in 2D CIPS arising from the scattered domains, instead of the fluctuation of sample thickness. We have added this new result in the supplementary information as **Fig. S11**. Further evidence can be found in the dependence of the zero-bias photocurrent mapping on poling voltage as shown in **Fig. S4**. By voltage poling, the scatter domain structure can be polarized into a single uniform domain, which influence the photocurrent accordingly. This is also a direct manifestation of ferroelectric domain effect, instead of the uneven thickness effect.

Figure R12: (a) Zero-bias photocurrent mapping with higher spatial resolution. (b) Photograph of the CIPS device. The yellow, blue, and red dotted lines represent the top graphene, CIPS, and bottom graphene, respectively. (c) The photocurrent distribution along the dotted line. (d) The AFM topography of the CIPS used in the device. Inset is the AFM line profile.

Question 3: Why the authors used the fitted data in Fig. 2a, but not use the experimental data?

Response: We used the fitted data to estimate the V_{oc} in our photovoltaic study. The reason is to avoid the reading bias induced depolarization effect. In conventional photovoltaics, the V_{oc} is directly determined from the I-V measurement by sweeping the reading bias in a wide range. However, for 2D materials, due to the ultrathin film thickness at atomic scale, low voltage applied in the direction perpendicular to the 2D plane results in large electric field strength. For example, for films with the thickness of 1 nm, applying 1 V DC voltage leads to an electric field strength at 1×10^7 V/cm in

the out-of-plane direction. This value is much larger than the coercive electric field strength of 2D ferroelectric CIPS ($\sim 7.5 \times 10^5$ V/cm as shown in **Fig. S2**). Therefore, applying wide range reading bias in photocurrent study leads to strong depolarization effect in CIPS. To avoid this problem, when measuring the I-V curve at bright state of our devices the reading voltage is therefore limited within the range of ± 0.1 V (equivalent to an electric field strength at $\sim 10^5$ V/cm). The V_{oc} is then determined through a linear fit to the I-V curve with extended DC bias range (see **Fig. 2a** and **Fig. R13**).

Figure R13: The depolarization effect of the reading bias in BPVE measurement. (a) The optical image of device #6. The blue dotted lines indicate the area of the top and bottom graphene electrodes. (b) The photovoltaic characteristic measured in a wide range reading voltage from -0.7 V to 0.3 V. The I-V curve deviates from linear when the reading voltage is larger than -0.3 V. By linear fit (red dashed line) to the I-V curve, we get a larger V_{oc} at -1.04 V (c) The V_{oc} dependence on poling voltage.

Question 4: The open circuit voltage data over different temperature is very scattered, as shown in Figure 3a, especially for the data near 310 K. More data are needed.

Response: We thank the referee for this suggestion. Following this suggestion, we investigate the temperature dependence of I_{sc} and SHG with a higher temperature resolution near 310 K. As shown in **Fig. R14a** and **R14b**, we find similar dependence relation on temperature for the I_{sc} and V_{oc} . The results are consistent with that presented in **Fig. 3a**. The ferroelectric to paraelectric phase transition temperature T_c of CIPS is further confirmed by the temperature-dependent SHG measurement. For all the sample thickness, we find that the T_c is at about 320 K. Our results are consistent with the results in *Nature Communications* 7, 12357 (2016). And for ultrathin CIPS

with 6 nm thickness, the temperature-dependent relations of I_{sc} , V_{oc} , and SHG near 310 K are almost the same. In the main text, we have updated the corresponding figures with these new results, while keeping our major conclusions unchanged.

Figure R14: Temperature-dependent (a) zero-bias photocurrent and (b) open-circuit voltage of CIPS device. (c) SHG measurements as a function of temperature for samples of bulk material, thick film (60 nm), and ultrathin film (6 nm). In the ultrathin film, the transition is not smooth probably from the finite size effect, which is consistent with the results in *Nat. Commun.* 7,1357 (2016).

Question 5: The thickness of CIPS presented in Figure 5b are shown in Fig. S8, but those height lines can't be used as the evidence. The authors should provide the detailed AFM images.

Response: We thank the referee for this suggestion. The detailed AFM images of all the samples used in **Fig. 5b** are now summarized in **Fig. R15**. And we have added these results in the supplementary information as **Fig. S8**.

Figure R15: (a-i) AFM image with height profile of the CIPS samples used in **Fig. 5b** with film thickness ranging from 7 nm to 230 nm. The scale bar is about several μm subject to the different sample sizes.

Question 6: For the devices, the thickness of the crystal is not uniform, such as in Fig. 1b, Fig. S4a, which can cause different current distribution in the device.

Response: The question is similar to *Question 2* that whether the distribution of photocurrent is attributed to the non-uniformity of the sample thickness in the device. Firstly, this possibility is excluded in this work that there is no obvious relation between the sample topography and the spatial distribution of the observed photocurrent (see **Fig. 1b** and **1c**). Secondly, the photocurrent distribution in the devices can be well tuned by an external field and temperature, which are the nature of ferroelectrics. Furthermore, similar feature have also been observed in a device in which the CIPS sample is with a uniform thickness (see **Fig. R16**).

Figure R16: (a) Zero-bias photocurrent mapping with higher spatial resolution. (b) Photograph of the CIPS device. The yellow, blue, and red dotted line represent the top graphene, CIPS, and bottom graphene respectively. (c) The photocurrent distribution along the dotted line. (d) The AFM topography of the CIPS used in the device. Inset is the AFM line profile.

Question 7: Some small concerns. (1) The scale bars should be added into the optical images, such as Fig. 1b, Fig. S4a, Fig. S6a, Fig. S7a and b. (2) The authors used the Keysight source meter B2902A to carry out the I-V measurement, the resolution of normal B2902A is 100 fA. The authors achieved a 2 pA level measurement, how did the author achieve electromagnetic shielding?

Response: We have added the scale bars in all the optical images in the main text and the supplementary information. For the issue on the resolution of the I-V measurements, the source meter used is Keysight B2912A with a current resolution of 10 fA. The statement of using Keysight B2902A in Methods is due to typos. We have updated this information in the revised manuscript. For the electromagnetic shielding, we used a tri-axial cable for electrostatic screening. And all the photocurrent

measurement is carried out in a well-grounded vacuum chamber. Under this condition, we achieved pA level I-V measurement in our study.

Point-by-Point response to review report #4

We thank this referee for his/her positive comments that “The topic is of current interest and the results are well presented. The manuscript can be considered for publication provided that several questions can be better addressed.” As detailed below, we have addressed all the issues raised by the referee and have revised the manuscript accordingly.

Question 1: How could the spatial confinement exclude the origin of I_{sc} from the Schottky barrier considering the vertical structure?

Response: This is a critical issue that needs to be excluded in this manuscript. As shown in **Fig. R17a** and **R17c**, a zero-bias photocurrent mapping with a higher spatial resolution, the photocurrent at the boundary of the overlapping area vanished sharply with an ultrahigh resolution defined by the scanning step at around 80 nm other than the laser spot size at $\sim 2 \mu\text{m}$. Furthermore, within the overlapping area, we find clear domain walls with negligible photocurrent (see the white lines in **Fig. R17a**). These observations could exclude the Schottky barrier effect because it is in contrast to the physics of the Schottky barrier in WS_2 and the metal electrode interface, in which the photocurrent decreased slowly in a wide spatical range of about 1-2 μm as determined by the laser spot size. In our photovoltaic devices, the photocurrent spatially decreased to the noise level within 100 - 200 nm, which is determined by the scanning step.

Moreover, all the other observations are consistent with this picture.

1. The zero-bias photocurrent is spontaneously positive and negative distributed without the dependence with sample thickness; however, it depends strongly on the polarization direction of the ferroelectricity.
2. The photovoltaic effect vanished when the temperature is above the critical temperature at about 320 K, above which the CIPS recovers the inversion symmetry. If it was related to the Schottky barrier, then no such sharp transition as a function of

temperature should be observed.

3. The photocurrent can be tuned by an external electric field, which is a typical mechanism of ferroelectricity, but not the physics of the Schottky barrier.

To conclude, all the observations are consistent or the same as the mechanism of BPVE, instead of the Schottky barrier, in all our samples with different sizes and thicknesses.

Figure R17: (a) Zero-bias photocurrent mapping with higher spatial resolution. (b) Photograph of the CIPS device. The yellow, blue, and red dotted line represent the top graphene, CIPS, and bottom graphene respectively. (c) The photocurrent distribution along the dotted line. (d) The AFM topography of the CIPS used in the device. Inset is the AFM line profile.

Question 2: Did the CIPS materials with different thicknesses have the same bandgap of ~ 2.9 eV, did they see any thickness dependence in the studied range? By the way, the quoted references gave the values of 2.4 and 2.9 eV for thicker or bulk samples.

Response: Following this question, we have checked the band gap width of bulk CIPS and thin films by the absorption spectrum measurement. As shown in **Fig. R18**, we have measured the absorption spectra for samples with thickness ranging from 10 to 100 nm. The band gap width of bulk CIPS with thickness greater than 100 nm is estimated to be ~ 2.87 eV by fitting the absorption edge with a linear function. For CIPS thin film with 10 nm thickness, the band gap width is determined to be ~ 2.83 eV, which is almost the same with the value found for the bulk material. We have added these results in the supplementary information as **Fig. S12**.

Figure R18: The absorption spectrum of CIPS on transparent PDMS substrate. A white light source is used as the incident light. The optical absorption coefficient (α), defined as $\alpha = \ln \frac{(1-R)^2 + \sqrt{(1-R)^4 + 4T^2R^2}}{2T}$. The inset is Tauc plot of $(\alpha hv)^2$ vs hv . The linear fitting near the absorption edge yields the energy gap width.

Question 3: Generally, for the BPVE, the V_{oc} varies linearly with the thickness d . This correlation could be provided.

Response: We thank the referee for pointing out this relation. Following this suggestion, we figure out the V_{oc} from devices with different sample thicknesses as shown in **Fig. R19**. However, other than the linear dependence on thickness, the V_{oc} show size effect that V_{oc} is lowered significantly as the thickness of CIPS is above 80 nm. When the CIPS thickness is less than 80 nm, the V_{oc} is not obviously dependent on thickness. According to the BPVE theory, the V_{oc} is dependent on film thickness

and the built-in electric field (E_{bi}). Nevertheless, in CIPS based BPVE devices, the E_{bi} is influenced by the ferroelectric polarization, graphene thickness, and the CIPS thickness. These mixing factors lead to the less obvious dependence of V_{oc} on CIPS thickness.

Figure R19: Open-circuit photovoltage of CIPS devices with various thicknesses. When the thickness is less than the free path length, the V_{oc} is independent of the thickness; however, when the thickness exceeds the free path length, V_{oc} quickly drops to zero. This observation is consistent with the BPVE theory, which is independent of sample thickness.

Question 4: In Fig 5b, the normalized I_{sc} values have a sharp drop when the thickness is larger than 80 nm. Does this observation mean the generated carriers collected by the electrodes have a sudden drop when the thickness is larger than 80 nm? Why a sudden change instead of a smooth one is observed, more physical explanation could be provided.

Response: This is a common feature in all photovoltaic effect, including the studies of solar cells. We illustrate this result using the cartoon in **Fig. R20**. When the separation between the two electrodes in the device is smaller than the free path length (in some of the literature, it is also termed as carrier diffusion length), the electron and hole created by absorption of one photon can be driven to the two electrodes, yielding a nonzero net current. However, when the separation is larger than the free path length, only the electron and hole can diffuse to the electrodes, yielding zero current. For this reason, in solar cells, the separation distance of the collecting electrodes should be finite. However, for practical applications, it should be large enough. Otherwise, the absorption of light is low, yielding low quantum yield. In generally, different materials have different free path length, depending strongly on a lot of factors, such as carrier lifetimes and mobility edges. The indirect band gap

materials are likely to have longer lifetimes, hence when integrated in photovoltaics they allow large electrode separation. In contrast, the direct semiconductors such as perovskites, CIPS, β -Ga₂O₃, In₂O₃, and SnO₂ etc, the free path length is typically about tens of nanometers to hundreds of nanometers. Our results are consistent with the previous reports, which can be found in the study in BTO thin films (*Phys. Rev. Lett.* 118, 096601 (2017) and *Phys. Rev. B.* 90, 161409(R) (2014)).

Figure R20: Illustration of the diffusion process in photocarriers created by irradiation. The electron and hole will be driven by E_{bi} . The created electron and hole can reach the two electrodes when the distance is less than $2l_0$, where l_0 is the carrier diffusion length. Here we assume the electron and hole have the same diffusion length. When $d > 2l_0$, the created electron and hole can not reach the electrodes efficiently, yielding sudden drops of photocurrent.

Question 5: As known, the BPVE will generate a large photocurrent density in small-thickness materials with a relatively low open-circuit voltage. Considering the practical application, is there any method to overcome this shortcoming?

Response: We strongly agree with the referee on this comment that a large photocurrent density often exist in small-thickness materials, which is the main finding in this manuscript. To enlarge the open-circuit voltage in the BPVE device, a tandem strategy can be utilized (*Nat. Photon.* 5, 672–676 (2011)). Considering that the 2D CIPS thin layers are transparent, we can make several CIPS cell stacking in series. As shown in **Fig. R21**, the tandem configuration of two CIPS based BPVE devices can produce the superimposed open-circuit voltage while keeping the photocurrent intensity unchanged.

Figure R21: Schematic of the tandem photovoltaic cells composed of two CIPS based BPVE devices. The ideal photovoltage of the tandem photovoltaic cells is the superposition of the output photovoltage in each device with $V_1 + V_2$.

Question 6: Recent reviews on 2D Ferroelectrics, such as *Adv. Electron. Mater.* 2020, 6, 1900818; *Adv. Mater.* 2021, 33, 2005098, could be included, which might be useful to the broad readership to better understand this emerging topic.

Response: We thank the referee for the comment. We have added these valuable references in the revised manuscript (see **Ref. 22** and **Ref. 23**).

Question 7: The manuscript is in general well written. However, a few typos should be corrected, such as Page 2 “The enhanced BPVE is associated to”.

Response: We have improved the presentation of the whole manuscript and have modified all the typos in the main text and the supplementary information.

D: Major changes to this manuscript

1. We have fixed all the typos in the manuscript. Changes made to the text in the manuscript and the supplementary information are marked in red. We have added scale bars in all the figures, if necessary.
2. Some important changes are made to **Table I**. We add the S-Q efficiency and the power conversion efficiency in the last two columns of **Table I**.
3. The power conversion efficiency calculation of CIPS device is added in the supplementary information as **Supplementary Note II**.
4. In the section “**The dimensionality effect of the BPVE**” in the main text, we have discussed the mechanism for the dimensionality effect of BPVE with a few sentences. The sentences are “The observed dimensionality effect in BPVE photovoltaics can be intuitively interpreted from two aspects. Firstly, it is well-known that the separation between the two collecting electrodes in a photovoltaic device should be smaller than the free path length for efficient photoelectric conversion (see the Cartoon in Fig. S13). With a capacitor-like device structure, the atomic thickness of 2D CIPS naturally fulfills this prerequisite. Secondly, it has been pointed out that when the dimensionality is reduced the BPVE is more efficient with enhanced shift current⁵¹, which may originate from the Berry connection of the Bloch functions⁵².”.
5. Five new references in the reference list as Ref. 22, Ref. 23, Ref. 50, Ref. 51, and Ref. 52. The journal names of the references are changed to Nature Communications style..
6. A revised supplementary material is prepared, providing more supporting data for the major conclusions in the main text. We outline the major results in the supplementary material as following.
 - a) We add the detailed AFM images for all the CIPS samples summarized in **Fig. 5b**. These results are now summarized in **Fig. S8**.
 - b) We add the wavelength-dependent BPVE of CIPS in the supplementary information as **Fig. S9**, showing that the BPVE is disappeared when the wavelength of the exciting laser is smaller than the band gap of CIPS. The band gap width of CIPS is estimated to be around 2.8 eV.
 - c) We add the V_{oc} as a function of sample thickness in the supplementary information as **Fig. S10**, which also shows a similar thickness effect like the I_{sc}

dependence on the sample thickness.

- d) We add the high resolution zero-bias photocurrent mapping of a new CIPS device in the supplementary information as **Fig. S11**. The spatial scanning step is 80 nm.
- e) We add the absorption spectrum of different thicknesses CIPS in the supplementary information as **Fig. S12**.
- f) We add a schematic cartoon to clarify the free path length of BPVE in the supplementary information as **Fig. S13**. This picture shows that when the separation of collecting electrodes is larger than the free path length, the BPVE induced photocurrent disappears quickly.
- g) We add the transport character of all the devices with different thicknesses in the supplementary information as **Fig. S14**.
- h) We add the transmission spectra of 10 nm and bulk CIPS measured at 405 nm in the supplementary information as **Fig. S15**.

REVIEWERS' COMMENTS

Reviewer #1 (Remarks to the Author):

The authors have addressed all questions satisfactorily. The manuscript is acceptable in its current form.

Reviewer #2 (Remarks to the Author):

The author has revised the article according to the requirements and made great improvements. This manuscript can be accepted.

Reviewer #3 (Remarks to the Author):

The authors gave suitable revisions according to all my previous comments. The revised version can be published now.

Reviewer #4 (Remarks to the Author):

In this revised version, the authors have addressed all the questions properly. The revised manuscript can now be accepted.